# Class Barriers to Merit in the American Professoriate: An Archaeology Example and Proposals for Reform

**Michael J. Shott**

Department of Anthropology, University of Akron, Akron, OH 44325, USA; shott@uakron.edu

**Abstract:** Consumers and academics alike perceive a status hierarchy among American universities. By this perception, professors are placed in the status hierarchy befitting their scholarly merit. However, a recent study of the archaeology professoriate found no consistent correlation between faculty placement and merit. This essay identifies reasons for the lack of meritocracy, some unique to archaeology and others common to many fields. Archaeology, similar to the American academy at large, ignores class as a bias that handicaps some while favoring others. Notwithstanding challenges of definition and measurement, class should be treated equally with race, gender, and other biases in an academy's pursuit of true meritocracy.

**Keywords:** faculty; placement; merit; cumulative advantage; American universities

## 1. Introduction

America is a consumer society. Routinely, Americans make judgments about the relative merits of products that range from chewing gum to home mortgages. People know that brands or variants in any category of goods and services can be ranked from higher to lower, from better to worse. Brand X can be distinguished from the good stuff, so beware of cheap imitations; *caveat emptor*.

Surely, then, Americans are shrewd judges of quality in a wide range of commodities; we are savvy emptors who heed the caveat in the process of making informed choices. All the domains of commerce that seek Americans' money and custom encourage the belief, if not always the discipline, of independent data-collection and evaluation that underlie truly informed judgment. Indeed, the ways that sellers encourage the belief comprise a catalogue of ploys that range from happy talk to half-truths to outright deceptions; McDonald's, some may be surprised to learn even now, does not do it all for you. From air fresheners to autos, consumers are encouraged to confuse quality with celebrity endorsement and aggressive marketing, as though Michael Jordan's undeniable basketball talents vouchsafe his expertise in men's briefs and hot dogs. Being wary is necessary but insufficient in the pitilessly feral landscape of American commerce. A better watchword for that wilderness would be *emptor sapiens*: let the buyer be wise.

Being wise, or at least reasonably informed, presupposes the availability of evidence, knowledge of how and where to acquire it, and the ability to process it in order to independently—and legitimately—arrive at conclusions about the comparative merits of competing brands. Wise judgment also encompasses the possibility that differences among brands—in price, the celebrity of endorsers, and popularity, or perhaps even in valid measures of performance or quality—may be apparent more than real. In equal dosages, Tylenol© and generic acetaminophen are exactly the same thing despite any difference in price or brand appeal.

However much it cultivates the image of an ethereal realm set apart from the profane world of commerce, what is true of products from chewing gum to washing machines is equally true of higher education. For one thing, no one who has tracked tuition charges over the past 40 years can doubt that universities, especially private ones, are avid seekers

of money and custom. However, more importantly, American higher education, again mostly its private varieties, claims distinction and asserts a clear, unambiguous hierarchy of quality, an educational Great Chain of Being.

Accepting these claims, American consumers know that there are best schools, good schools, and then the rest. They also know what comprises good schools and lesser ones, applying a set of criteria that includes presumed measures of quality of input (viz., undergraduate students; Americans exhibit much less interest in graduate education and its varieties) such as mean aptitude-test scores and acceptance rates, of processes such as the presumed quality of the faculty, however judged (and, often, of things such as climbing walls and the thread-count of residence-hall bed linens), and of outputs such as student–customer satisfaction, placements, and career earnings. Institutions thus comprise a hierarchy that corresponds to the perceived native ability of those who populate them at levels from undergraduates to senior faculty. Like silts in suspension, the cruder among them settle out first, the progressively finer do so higher and higher above them. Each stratum in the column is relatively homogeneous, distinguished from those just above or below it in slight degree and from those farther distant in starkly categorical terms. If you are among the best, you attend or teach at the best schools. If you are merely good, then your province as student or professor is good schools, and so on. There is a corollary to this iron law of status judgments, from student to professor. In this moral universe, affiliation is determined entirely by intrinsic merit and that affiliation is destiny; where you are is where you deserve to be.

This is what consumers believe. One might suppose that academic insiders—professors themselves—would know better, or at least different. However, surprisingly many professors draw inferences from their own and colleagues' affiliations that are no less blithe or tendentious than are consumers' [1]. To many academics, therefore, and regardless of one's comparative scholarly record, "where one works is a marker of status" [2], see also [3] (pp. 107–108), and many faculties in what it pleases consumers to call "prestigious" universities "are understandably invested in their schools' rankings" [4] (p. 6). In this world, "Positions in top universities…have a double significance, symbolic and instrumental" [5] (p. 245). Thus, the scholars who are fortunately situated at one or another exalted university enjoy both the symbolic capital and prestige of their affiliation and the material benefits— ever growing, at least in private universities [6]—in salary, workload, research support, and setting that attend it [7]. Conversely, "it is often difficult for academics at less distinguished campuses to receive appropriate recognition for their achievements" [4] (p. 13).

This is because guilt-by-affiliation logic, by consumers or academics, projects to individual scholars the judgments made of their institutions. And, when your institutions lie on the Great Chain's lower links, that collective judgment considers them "'weaker universities", "less prestigious universities", "poor-boy schools", or "unproductive universities" [8] (p. 342). By extension, it applies the same adjectives to you. In this way, what sociologists-of-science call "cumulative advantage" (institutional variation in support for scholarship that largely correlates with university and department size and the highest degree awarded by the department) and its guilt-by-affiliation handmaiden deliver a double bonus to the fortunate and a double whammy to the rest. Like affluence for the bourgeois in Weber's formulation, in the academy affiliation "provides the theodicy of good fortune for those who are fortunate" [9] (p. 271). The brutal corollary of guilt-or-glory logic for professors of modest placement: you are where you are because that is where you deserve to be.

A tangled mess of misconceptions underlies this logic. Among them, leave aside the shamefully transactional nature of "elite" college admissions whose mere iceberg-tip the Varsity Blues scandal revealed. Leave aside the associated belief that higher education involves a sort of osmotic process that occurs naturally, without conscious effort and no matter any student's attitude toward learning. In this view, even those indifferent to learning—or worse, to judge from accounts of student conducted at "prestigious" private universities, e.g., [10]—receive educations of a quality commensurate with their institution's perceived stature. A heavy-drinking über-slacker at an exalted university who attends

class only twice a month will emerge well-educated despite himself. Conversely, even the most dedicated student at a mediocre university receives an indifferent education. Leave aside the abundant evidence that, even among those students admitted for ostensible merit, social and financial capital correlate significantly with the merit measures they can boast and that the institutions that admit them trumpet. Leave aside what can only be called the ludicrous risibility of ranking schemes. Leave aside as well the reality that, so far as it concerns the quality of higher education *qua* education—as opposed to affiliation branding and social experience and connections—*where* one attends college matters much less than *how* one does.

Leave all of it aside. This essay's subject is the professoriate and how well the American faith in meritocracy, in apparently shrewd judgments of quality, explains which professors are placed where. *Placement* denotes university affiliation with the status, if any, it delivers. Ivies deliver high status, whether earned or not, Apache Creek Junior College practically none. *Meritocracy* denotes the widely held ethos that, in academic context, scholarly ability and record should determine placement. Certainly, the widespread if unexamined belief is that professors are placed according to merit in this most meritocratic of all possible worlds. This essay, with related work that explores the matter in more detail [11], examines this faith from the perspective of an archaeologist of working-class origins who graduated from a no-account state university and labored for decades in the lower ranks of the academy's Great Chain despite a reasonably productive scholarly record [12]. *Emptor sapiens*; the essay interrogates the extent to which placement and merit coincide. If they do not, what factors explain the divergence?

When American consumers so much as contemplate such questions, they are apt to answer them by assumption. Whatever your scholarly record and whatever your standing in your field, shrewd American consumers judge you by your affiliation. If you are at an exalted university, then you must be eminent. If, however, you labor at the sort of school that advertises on billboards, then you cannot be any good. If you are at Nowhere U, that is, then you must be a nobody. The view also takes for granted free faculty mobility governed solely by merit, such that each increment of earned merit is rewarded at once by a proportional rise in affiliation status. If you do not rise, then you did nothing to merit rising. These views and the judgments on which they rest are of long standing, tracing back to James Cattell's equation of universities' eminence with the presumed eminence of the faculty who populated them [13] (p. 408), an equation that is legitimate only when faculty eminence is judged wisely. The prevailing views are correlates of the facile silts-in-suspension logic that consumers apply uncritically to higher education.

Over the past four decades, such ill-informed judgments of both students and faculty have only been reinforced by highly tendentious rankings, notably by *US News*. These are neither fair nor balanced in representing the complex realities of higher education, even from the consumer perspective that they profess to serve. Instead, they are branding exercises that start from their conclusions and rig the "evidence" they adduce (e.g., reputational surveys notoriously susceptible to gaming, rigid student–faculty ratios, and retention rates calculated in blithe indifference to the socioeconomic conditions that determine them) to support them. The rankings are designed to shape consumer preconceptions about quality in higher education and then to reinforce the resulting misconceptions. In this way, "the Yales of the world will always succeed at the *U.S. News* rankings…[that are] designed to measure Yale-ness" [14] (p. 74).

> *"To the extent that science is universalistic [unbiased in placement], attainments should be allocated on the basis of contributions to scientific knowledge…reflected in publication productivity"*. [15] (p. 51)

## 2. Meritocracy in the Academy?

So, is the academy meritocratic for professors, as consumers suppose? This essay is drawn from a larger cross-sectional analysis of archaeologists in the American academy as of AY2016-17. The details of data collection, analysis, and inferences are found in [11].

(That study and [16] defended measures of scholarly productivity used and addressed prospective criticisms that factors besides scholarship [e.g., mentoring, administrative service] are both possible to gauge and fair to treat as alternatives.)

Briefly, the study found that at initial hire archaeologists did not sort by scholarly merit either between highest-degree program levels (BA, MA, and PhD) or within the perceived PhD-program hierarchy. Then, cumulative advantage in the practical absence of opportunities for placement mobility explained most later-career productivity, which overlapped greatly between degree-program levels. This world resembles an imaginary university whose faculty was populated by both callow undergraduates and distinguished senior scholars while some equally distinguished scholars labored instead at trade schools and grade schools. Far from corroborating consumers' faith in both natural hierarchies of quality and their ability to discriminate intelligently among varieties within them, the study suggested that faculty are similar to acetaminophen; being packaged as Tylenol$^{©}$ or Brand X is unrelated to intrinsic scholarly merit. Where archaeologists are situated on the Great Chain reveals nothing about their academic merit.

The larger study cited many sources that reached similar conclusions for other scholarly fields. Across the American academy, evidently, faculty placement and merit correlate poorly. As one veteran of the academic trenches and skeptic of notions of meritocracy put it, "When you look at which scholars end up where, there is nothing logical or fair about it..." [17] (p. 110). Conclusions so consistent beg an obvious question: if merit does not explain placement in the academy's Great Chain of Being, then what does?

### 3. Why There Is No Meritocracy in the American Academy

There are several explanations, again discussed in detail elsewhere [11,16]. Briefly, in archaeology, one is the strong preference for particular geographic areas of research, especially Latin American civilizations that raised the temples and pyramids that draw tourists [18]. Legitimate among many other geographic areas of much less glamorous appeal—not least eastern North America—these are the regions that the popular imagination, assisted by Indiana Jones, identifies with archaeology as romantic adventure. Those of us too dumb to choose such areas compete for proportionally fewer openings, no matter our scholarly merit relative to those who work in favored regions.

Broadly across American higher education, another explanation is simple chance in an academic labor market that, for decades, has produced PhDs far in excess of demand for their services; from his academic administrator's perspective, the pseudonymous Dean Dad limned the constriction of opportunity that such conditions impose:

> "*the relative ease of finding adjuncts for a given discipline actually mitigates against its getting a line* [a new hire]. *If you can only afford to hire one full-timer, and you have requests from both history and, say, pharmacy, what do you do? If good history adjuncts are easy to find, and good pharmacy adjuncts are nearly impossible, you give the line to pharmacy*". [19]

Whatever any historian's scholarly chops or how ordinary a pharmacy scholar might be, the latter wins the job for reasons unrelated to merit. Surveying a supposedly meritocratic German academy over a century ago, Weber observed about the prospects for advancement that "chance does not rule alone, but it rules to an unusually high degree. I know of hardly any career on earth where chance plays such a role" [9] (p. 132). Other sources [20] (p. 146), [21] echoed this judgment in explaining their authors' own career trajectories and others'.

However, chance is not placement's only explanation for the poor correlation that consumers assume between scholarly merit and placement. Prominent among other explanations are social pedigree and capital; class, in a word. Class may be difficult to define; it is an elusive "fluency which evades analysis" [22] (p. 9). Like pornography, though, it is comparatively easy to see or sense, if difficult to acknowledge. Social class affects faculty placement because "more prestigious, research-oriented institutions have drawn their professors disproportionately from the higher social strata" [23] (p. 323). That

observation was made over 40 years ago. Slightly later, "lower-class respondents with Ph.D. degrees from high-ranking universities were less likely than middle-class respondents with such degrees to obtain positions at prestigious universities" [24] (p. 1). For archaeology, a later study identified pedigree effects associated with the perceived prestige of professors' BA or PhD degrees [25] (see also [26] for sociology and [23] (Table 2) for the academy in general). A recent survey of British academic and preservation archaeologists found somewhat similar results [27] (Section 2.10). As it is today and has been for decades past, so it was a century ago when Weber [9], a scholar of not-inconsiderable stature, railed about such placement injustices among faculty in the German academy.

> "*American society always has had to come to grips with social conflict. A favored way of doing that has been to reconceive class divisions rather as ones involving race or other forms of inherent or ascribed or volitional identity...This has led, on the one hand, to great gains in recognizing the basic humanity of nonwhite people...It has also functioned as an evasion. Class in this context can be made to not matter*". [28] (p. 250)

> "*There was no language for what I represented on campus. Scholarships and student organizations existed to boost kids from disadvantaged groups such as racial minorities, international students, and the LGBTQ community. I was none of those things...*". [29] (p. 261)

## 4. The Invisibility of Class

Can class considerations in college admissions or recruitment to faculty ranks "be made to not matter," in Fraser's wording? *Made* to not matter? *Do* not matter, period. This attitude, too, is typically American: the blithe neglect of a glaringly obvious social fact whose acknowledgment, let alone any effort to remedy its unfair advantages, is regarded as an offense against politesse. In Britain and France, by contrast, social class is very nearly an obsession, defined and mapped as assiduously there as it is studiously ignored here.

One longitudinal study of educational attainment concluded that "The chances of a high ability student obtaining graduate or professional education, where ability considerations would be presumed to be determinant, are approximately 3.5 times better if he comes from a family with high socioeconomic status than from a low socioeconomic status family" [30] (p. 853). Unpack this turgid statement, as postmodernists might say. The authors implied that access to and success in graduate education is presumed to be meritocratic. This *assumption* of meritocracy—not its reality—seems valid, as a social fact. Yet, their own evidence revealed that, despite the blithe assumption of equal access and meritocratic outcomes, success depended more on class advantage than any individual merit brought to the case. This longitudinal study reached only as far as *access* to education, not also to post-graduate placement, but it is no great leap—indeed, more nearly a baby step—to that further conclusion [31].

Even now, therefore, "the professoriate is, and has remained, accessible mainly to the socioeconomically [and academically] privileged" [32] (p. 45). Even when the unwashed manage to grasp the brass ring of academic appointment, too often their prize turns out to be made of tin foil. Thus, the Carnegie Research I and II institutions disproportionately employ faculty whose own parents were very highly educated, whereas comprehensive universities—rough functional equivalents to France's common universities and Britain's polytechnics—and much lower on the Great Chain, disproportionately employ faculty who were themselves first-generation college graduates [33] (p. 62). These patterns amount to an academic-inheritance system that preserves privilege across generations. Inequities so glaring and historically persistent have inevitable consequences. Thus, a scholar's work may be good "but if he is too old, or too young, or located in the minor leagues, it will not be recognized...and will not bring him the professional advancement which he could claim if he were of the proper age and located at the proper university" [3] (pp. 128–129).

From Weber to today, then, class bias has been obvious to some in the academy. Yet, it is strangely invisible to most. To them, exalted affiliation is plain evidence of earned, eminent status rather than social advantage. Such logic may comfort the comfortable.

Surely some who hold exalted statuses worked hard to attain it, but others are disinclined to acknowledge the role of chance or unearned advantage in theirs. In Posselt's experience, many "downplayed how socioeconomic status is related to the attainment of elite academic credentials…[thus] reframing prestige as merit" [34] (p. 103); this too is Weber's theodicy. This academic equivalent to vulgar prosperity theology has spawned its own apologetics [35], which seek to demonstrate not only the essential rightness of unearned privilege but also how this inequity somehow is preordained, inevitable, and for the best in this best of all possible worlds.

Today, then, the academy freely recognizes some inequities and their sources, yet is remarkably obtuse about others. In particular, it continues to ignore its own class bias and the corresponding advantage that many of its occupants possess. A recent study, for instance, analyzed the hiring networks of major PhD programs for American archaeology's encompassing discipline of anthropology. The authors proudly proclaimed anthropology's undeniably commendable "commitment to fighting social inequalities" [36] (p. 2) and repeatedly noted the legitimate need to consider "gender, race, ethnicity, faith, age, sexual orientation, and dis/ability" [36] (p. 3). Conspicuous by its absence in this list of ills—in a piece whose title included the word "inequalities" and whose catalogue of disadvantaged statuses ran to seven items—was any concern for social class.

This is no surprise. Notwithstanding its rhetorical commitment to equity, anthropology is an elitist discipline whose professoriate is no better than others, perhaps worse than some, in being drawn from the socioeconomically elite and having much higher probabilities of parentage of high academic accomplishment than the population at large [32] (Table 4). Much the same is true elsewhere, for instance in history, where categorical information on job candidates and hires collected both by hiring institutions and by the discipline at large in efforts to eliminate bias nevertheless "demonstrate a staggering indifference to class" [37] as a bias to overcome. Such neglect of class advantage may be because "With education and the progression through professorial ranks…many middle- and upper-class scholars may not wish to consider the ways in which their achievements are facilitated by class privilege" [37]. As a result, archaeology within anthropology—really, the academy at large—is blithely indifferent to the social-class advantage that many of its practitioners enjoy and to the obstacles thrown in the path of lower-class colleagues. This is because, in the United States, "the articulation of the relationships between class, race, and gender are simply evaded" [38] (p. 228).

> "the academy, in an era when every aspect of identity is scrutinized and revealed as 'socially constructed,' has accepted with manifest complicity the workings of class differentiation. It has turned two thirds of its mantra, 'race, sex, and class,' on itself…But class? How can the professoriate expose the class biases of its operating assumptions without dismantling the academy as we know it? We live these biases. We assume that Harvard professors…are more intelligent, more insightful, and deserving of greater admiration…". [39] (pp. 26–27)

## 5. Creating True Meritocracy

Well, some of us talk about class and decline to esteem placement uncritically. Some, that is, grant the abilities of professors at any university and gladly extend them the admiration we think they merit. However, we do this only upon familiarity with their scholarly records, not as leaps of faith from the accidents of their affiliations. Not all of us, that is, think like *US News* and its eager consumers. Undeniably, though, even today the academy at large ignores class advantage and its effects in so much as undergraduate admissions, e.g., [29], certainly in faculty hiring or career advancement. Still, we live these biases.

As a result, in the United States, faculty recruitment incorporates legitimate race and gender preferences to overcome historical barriers, e.g., [40], and archaeology is engaging more constructively with native peoples [41]. Yet, hiring efforts that so much as acknowledge class bias, let alone make efforts to redress it, are exceedingly rare if any

exist. My nearly four decades' experience of reading job advertisements and serving on search committees recalls not a single example. Today, even documenting the changing demographic profile of the professoriate routinely takes into account race and gender [42] yet remains utterly silent on social class [cf. 27 for a recent exception concerning the British profession].

We are long past time for change. At all stages from undergraduate to graduate admissions to new faculty and mid-career hires, recruitment must consider social class among the criteria that guide selection. With respect to faculty hires, that is, the boilerplate of race, gender identity, and sexual orientation must be expanded to encompass social class. As an essential step toward true meritocracy, we must handicap for class advantage and its absence.

Undeniably, though, such affirmative action for working-class scholars [43] will prove a tough sell. There is the problem of empathy fatigue, American society already having wracked itself, highly imperfectly and with growing backlash in recent years, to counter-balance generations of racial and gender discrimination. There are corollary problems of definition (What makes one working-class?) and measurement (On whatever scale, what is the dividing line between working-class and higher classes?), which, granting the fluency or elusiveness of class, could be resolved with good-faith effort and care (e.g., measures of socioeconomic status, parental education level). Then, there is the question of how and how much to favor or promote working-class scholars compared to others. Finally, in a world where "the privileged *are* the meritorious" [44] (p. 554), not as a matter of native ability but the cultural capital that enables them to define as abstract "merit" the characteristics and abilities they possess disproportionately, what constitutes merit and its measurement require reconsideration. The work needed will range from undergraduate SAT scores that reflect socioeconomic status more than native wit to measures of faculty scholarship that sort the truly productive from the rest.

## 6. Conclusions

In a world crowded with injustices, the placement inequities of university faculty are modest in perspective. So yes, work then without disputing, as someone once recommended. Many professors of low placement but higher merit do. To a point. Yet, even a century ago, that point exceeded Weber's capacity to forbear when contemplating inequities in a German academy that, in his judgment, few productive scholars "could endure...without coming to grief" [9] (p. 134). As in Germany then, so in the United States for generations since, there have arisen many reasons for such grief. Guilt-or-glory by stupid accidents of affiliation allow false and idle impositions of reputation to pervade American higher education.

However, after a century or more of placement inequity, the academy and American consumers equally still fail to so much as acknowledge, let alone address, the problem. Whatever this failure implies about academics' views of their own placement and their colleagues', it has permitted consumers to assume a correlation between placement and scholarly record—a complacent guilt- or glory-by-affiliation view unchallenged by evidence or reason. Reputation as a matter purely of affiliation surely justified Weber's exasperation a century ago and others' today. Whether it continues to beset future generations is a question that deserves the academy's full attention, not its neglect.

Looking back in anger had its day, whatever good it may have achieved. Now is the time to look ahead, in any state of mind, toward a higher-education system that, at levels from undergraduate admission to faculty hiring and advancement, is open to all equally whatever their class status. Consumer misconceptions of quality in education are deeply rooted, practically impervious to questioning. *Caveat corrector*: reformer beware, they will not change. When our ethos of academic opportunity allows true merit to triumph, at least then consumer misconceptions might finally align with practice. Come that millennium, the academy will approximate the meritocracy that it loudly proclaims today.

**Funding:** This research received no external funding.

**Institutional Review Board Statement:** Not applicable.

**Informed Consent Statement:** Not applicable.

**Data Availability Statement:** Not applicable.

**Acknowledgments:** Cheryl Claassen guided this paper through the editorial process. Several anonymous reviewers made constructive suggestions. Any omissions or errors are my fault.

**Conflicts of Interest:** The author declares no conflict of interest.

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
