# Peer review of "Class Barriers to Merit in the American Professoriate: An Archaeology Example and Proposals for Reform"

_humans, doi:10.3390/humans3010001_

Round 1

Reviewer 1 Report

It is a high-quality paper to study "archaeological sociology". I have a suggestion for your further studies. Perhaps you need to do a study with historical perspective -- to clarify the diachronic change of archaeologists against background of American society. This paper more focused on the contemporary statues. I am thinking whether the New Archaeology movement played a role? What about Processual-plus or Post-processualism? 

For this paper, please address your empirical data if possible, as a table or an appendix. 

Author Response

Responses in green italics.

REVIEWER 1

t is a high-quality paper to study "archaeological sociology". I have a suggestion for your further studies. Perhaps you need to do a study with historical perspective -- to clarify the diachronic change of archaeologists against background of American society. This paper more focused on the contemporary statues. I am thinking whether the New Archaeology movement played a role? What about Processual-plus or Post-processualism? 

 For this paper, please address your empirical data if possible, as a table or an appendix. 

I’m sincerely glad that Rev. 1 likes this “high-quality” paper.  S/he makes constructive suggestions about how to focus or structure later studies historically, which I’ll consider when returning to the longer ms., not this essay.  That is, the comments do not directly address any flaws, omissions or questionable points of logic in the ms. under review.  Therefore, there are no specific responses required.  The comment on “Processual-plus or Post-processualism” (actually, merely an archaeological expression of postmodernism that does not deserve a unique moniker) is unclear to me.  I don’t understand how I can take it into account in any future study.  So far as it concerns my subject, my sense is that processualists (plus or original) are as elitist as professionals as postmodernists are.  But I’ve not collected data to determine any differences among them, leaving aside the question of how to classify archaeologists by their kind or degree of processualism, or really, modernism vs. postmodernism.

As discussed with the issue editor, I cannot “address” empirical data from the cited source.  It would not nearly fit in a single table, and an appendix would amount to duplicating the published source.

Rev. 1 answered “yes” to every question except about “empirical results presented clearly?, which s/he answered as “can be improved.”  No empirical results were reported; again, this is an essay.

Reviewer 2 Report

Thanks for the opportunity to review “Class Barriers to Merit in the American Professoriate: An Archaeology Case Study and Proposals for Reform”. As the title indicates the author makes the case that the perceived status hierarchy in American universities is not matched by scholarly merit and that there is an elitist bias in hiring practices that privileges those from more affluent socio-economic backgrounds. While the author presents a strongly worded argument, there are some wider considerations that the author might find useful particularly in considering reform proposals.

The author refers to American universities meaning universities in the USA.

Part of the argument made in the paper refers to university rankings. Reference is made to the U.S. News university rankings but there are a number of international ranking schemes (e.g. THE, QS etc.) with published methods for evaluation. There are also national schemes (although not in the USA) that evaluate research (e.g. PBRF, REF, ERA) as well local assessments by universities of qualifications from overseas universities. The point being, that there are a number of alternatives that the author might want to highlight that would better represent university quality. Mention of these might be considered since they could bring a degree of objectivity to university rankings.

In colonialist societies, the impact of past social injustices had marked economic impacts on indigenous peoples visible today in such things as health status, longevity, and participation in universities (along with many other areas). The equity literature when considering indigenous issues but also of other marginalised groups refers to intersectionality, that is the ways in which different aspects of a person's identity can expose them to overlapping and compounding forms of discrimination and marginalisation. This might be a productive avenue for the author to explore given the “invisibility of class” the author notes. Intersectionality is one way that socio-economic status (including its origins) might be considered by universities.

Universities that I am familiar with have equity policies and implementation plans. I do not know the USA university scene well enough to know whether there are equivalents, but it is in these policies and plans that I imagine issues of intersectionality would be identified and corrected.

Author Response

submitted via attachment

Reviewer 3 Report

I'm very pleasantly surprised by this paper and I'm glad that after decades of attention being reserved to gender/race inequities, we're finally talking about class in archaeology. Not only have the lower-classes been ignored, they have been demonized and dismissed in archaeology under terms like "populism". In other cases, poverty has simply been romanticized as something that only exists in exotic and faraway places like India and Africa. 

My own research, yet unplublished, focuses on the same class inequities from an European perspective. While I do not have anything significant to add to the author's research, there are nevertheless some perspectives I would like to share.

First, there is an entire historical and cultural world that could have been addressed. For instance, the author addresses commodities and it would have been interesting to see some Marxist perspectives on this. Chapter 1 of Marx's "Capital" is dedicated to commodities. In a way, what the author describes in the beginning of his text is a recent process in commodification, where people build their identities based on the products they consume, including the academic institution they attend. While this was reserved for elites in the past, this has been opened up to the middle-class (or what Marx called the Petty Bourgeousie). 

On a similar note, there has been fantastic work on how economics have changed in recent years, and how this has affected our perception of class. Top of the list would be Ernest Mandel's "Late Capitalism". Jameson's "Postmodernism", Eagleton's "Illusions of Postmodernism", Callinicos', "Against Postmodernism", and Stuarts "Everything, Everywhere, all the Time". all these books address, to some extent, the shift from class issues to gender/race issues.

Finally, there is some very interesting literature on class today. While Thompson's work on the English middle-class is very good, it is quite outdated. I would recommend checking out Selina Todd's "The People", but even more importantly, checking out Mike Savage's "Social Class in the 21st Century". This last book is particularly important because it shows how the tripartite class system has become more complex and how many people today have failed to recognize their own social position when it comes to class.

----------------------------------

Personally, I would have liked to see more discussion on networking. While hierarchies still play a strong role today, especially in old-school institutions like universities, the dominant metaphor of today's world is not the hierarchy but the open-network (see the dominance of Latour and Deleuze in the humanities). A big part of being successful today is not just the status signifier of universities like Cambridge or Stanford, and the economic capital required to access these, but also the social capital that these institutions provide. For instance, Cambridge holds several excavations in Greece (the European equivalent to Latin American archaeology), which cost upwards of 4000$ to attend per year. This, of course, is only accessible to a fraction of a percent of students. What these excavations then provide is access to data and to people, which allow these richer students to pursue their career further. This type of networking has been crucial in keeping jobs and funding restricted to the upper echelons.

Another aspect of hiring practices today, in archaeology, is favouring those who travel, even if temporarily, to another recognized institution. Once again, this is based on networking, or to use Bourdieu's terms, social capital. Like other classist assumptions, being able to travel requires capital that many lower-class archaeologists are simply unable to access. Furthermore, those who can most easily travel are those who work in the archaeological sciences, since scientific techniques can be applied universally across the globe. Once again, access to scientific training and laboratories is restricted, usually only available in richer countries (US, Northern Europe), and within these, only in richer universities.

Author Response

submitted via attachment

Reviewer 4 Report

I put my comments in post-it notes in the text.

Author Response

response submitted via attachment
